# Preclinical Models for Functional Precision Lung Cancer Research

**DOI:** 10.3390/cancers17010022

**Published:** 2024-12-25

**Authors:** Jie-Zeng Yu, Zsofia Kiss, Weijie Ma, Ruqiang Liang, Tianhong Li

**Affiliations:** 1Division of Hematology/Oncology, Department of Internal Medicine, University of California Davis School of Medicine, University of California Davis Comprehensive Cancer Center, Sacramento, CA 95817, USA; jiezenglily@gmail.com (J.-Z.Y.); zakiss@ucdavis.edu (Z.K.); weijie.ma@hitchcock.org (W.M.); rqliang@ucdavis.edu (R.L.); 2Department of Pathology and Laboratory Medicine, Dartmouth Hitchcock Medical Center, Geisel School of Medicine at Dartmouth, Lebanon, NH 03756, USA; 3Medical Service, Hematology/Oncology, Veterans Affairs Northern California Health Care System, Mather, CA 10535, USA

**Keywords:** lung cancer, precision oncology, preclinical models, patient-derived xenografts, lung cancer organoids, targeted therapy, immunotherapy, functional, review

## Abstract

The landscape of lung cancer diagnosis and treatment has transformed over the past two decades, ushering in the era of precision medicine for lung cancer. In systemic therapy alone, 43 drugs have gained FDA approval since 2020, encompassing chemotherapy, molecularly targeted therapies, immunotherapy, and antibody–drug conjugates. These therapies have significantly improved patient survival and quality of life. Improved preclinical models have been crucial in driving these advancements. Increasingly, the impact of tumor heterogeneity, along with its interplay with the tumor microenvironment and immune system, has been recognized as critical in influencing responses to these agents. This review provides an overview of the key characteristics, advantages, and limitations of current in vitro and in vivo models used in functional precision lung cancer research of systemic therapeutics.

## 1. Introduction

Lung cancer is the leading cause of cancer-related death worldwide [1]. Fortunately, the management of lung cancer has significantly evolved over the last decade, as reviewed in this special issue [2,3,4,5,6,7]. Lung cancer is a heterogeneous disease, and its clinical management depends on histopathological diagnosis, staging, and molecular and immune biomarker findings [8,9]. Non-small cell lung cancer (NSCLC) accounts for over 80% of lung cancer cases, followed by small cell lung cancer (SCLC), which makes up 10–15% [5]. It is reported that approximately 40% of newly diagnosed lung cancer patients exhibit metastasis disease [10]. Based on tumor origin, NSCLC is further classified into lung adenocarcinoma (LUAD), lung squamous cell carcinoma (LUSC), neuroendocrine tumors (~5%), sarcomas, and SMARCA4-deficient subtypes [11,12,13].

Precision oncology represents a paradigm shift toward a more individualized approach in cancer care, tailoring treatments based on the unique characteristics of each patient and their disease, especially for late-stage cancer patients [14]. NSCLC has become a prominent example of precision medicine among solid tumor cancers. All patients with NSCLC undergo immune biomarker PD-L1 immunohistochemistry (IHC) and tumor genomic profiling to guide the selection of first-line and subsequent systemic treatments, including molecularly targeted therapy or immune checkpoint inhibitors (ICIs), either alone or in combination with chemotherapy [7]. However, despite initial responses, resistance inevitably develops. The advances in cancer treatment would not have been possible without preclinical experimental models and technological improvements that enable the investigation of various aspects of disease initiation, progression, and tumor responses. Furthermore, patient-derived tumor models are essential for elucidating the mechanisms of drug resistance, evaluating novel drug efficacy with or without current treatments, and identifying biomarkers of response for patient stratification to inform future personalized therapies. This comprehensive review aims to summarize the different in vitro and in vivo models (Figure 1), highlighting their advantages, limitations, and applications in advancing personalized medicine.

## 2. In Vitro Model

### 2.1. Human Lung Cancer Cell Line Models

Human lung cancer cell lines are one of the earliest preclinical models and have been utilized widely to gain a plethora of information on cancer initiation, progression, and metastasis on the molecular and cellular levels. The use of these cancer cell lines significantly advanced our understanding the molecular biology of lung cancer. These cancer cell lines have shown the ability to maintain the expression of the “hallmarks of cancer” [15] (except for angiogenesis that requires the presence of stromal tissues). These include driver mutations that have been identified and are essential for the malignant phenotype. The identification of these driver mutations would have been impossible without the use of these cancer cell lines where one can knockout or overexpress certain oncogenes or tumor suppressors to understand their function and role in the development and maintenance of cancer. For example, in vitro lung cancer cell lines allowed the identification of TP53 mutations. Their significance in lung cancer has provided an understanding on the relationship between copy number gains, other mutations, and mutant-allele-specific imbalance in cancers [16,17,18]. Identifying sites of allelic loss or gain in lung cancer cell lines that are more frequent was also crucial in understanding the common key players in lung cancer pathogenesis. For example, lung cancer cell lines were used to show that RB is crucial in small cell lung carcinoma (SCLC) [19]. RB, just like CDKN2A, acts as a cell cycle checkpoint. While CDKN2A is known to be inactivated in many tumors, interestingly, inactivating point mutations in RB are mainly limited to SCLC and bladder cancers. In addition, the concept that there was a mutually exclusive RB-cyclin–CDKN2A tumor suppressor pathway and that this pathway can be inactivated by either mutational or epigenetic alterations in various human cancers was discovered using lung cancer cell lines [20,21].

Recent WHO clarification defines four molecular subtypes of small cell lung carcinoma (SCLC), which has therapeutic implications for emerging precision oncology treatment [22]. These subtypes were defined based on the differential expression of four key transcriptional regulators: achaete-scute homolog 1 (ASCL1, also known as ASH1), neurogenic differentiation factor 1 (NeuroD1), yes-associated protein 1 (YAP1), and POU class 2 homeobox 3 (POU2F3) [22]. Where multiple transcription regulators are expressed, samples are classified by the expression level of the transcription regulator that has the greatest relative overall expression. Studies focusing on understanding the critical signaling pathways dominating in SCLC subtypes would allow the research community to develop more promising therapeutic targets. For example, recent data indicated that Delta-like protein 3 (DLL3), including an antibody–drug conjugate, a bispecific T cell engager, and a chimeric antigen receptor (CAR)-T cell construct, can target SCLC-A subtypes [23]. Unfortunately, despite many targeted agents that are effective in many hematological and solid tumors, their success in treating SCLC have been disappointing [24,25,26,27].

The use of these various NSCLC cell lines is far more common and has allowed researchers to identify a plethora of genes that have been shown to be important in lung cancer pathogenesis [28,29]. The gene TITF1, known to play a role as a master transcription factor required for the differentiation of the peripheral airway, has been shown to be frequently amplified in lung cancer tumors and cell lines and has been shown to be a lineage-specific oncogene [30,31]. Gaining an understanding that the activation mutation in the kinase domain of the epidermal growth factor receptor (EGFR) was in close correlation with the sensitivity/resistance to tyrosine kinase inhibitor-based drugs (TKIs) was achieved via studies conducted using lung cancer cell lines [32]. Subsequently, almost all important biological characterization of intrinsic and acquired resistance mechanisms in the EGFR mutations was performed using cell lines [33,34].

In summary, without these cancer cell lines, the findings we gained and successfully “translated” to clinical applications would have been severely hampered. These cell lines are relatively inexpensive, scalable, and widely available [35,36]. Over 300 NSCLC cell lines are available in cell line collections [37,38]. Most NSCLC cell lines are derived from LUAD, and fewer LUSC cell lines are available due to the difficulty of culturing them. Thus, although sequencing efforts have allowed the identification of mutational status for many NSCLCs, unfortunately, LUSC cell lines tend to be less well characterized. Fortunately, these NSCLC cell lines have been shown to maintain some of the fundamental features of the tumors from which they were derived. Table 1 summarizes common human lung cancer cell lines and their genetic mutations. For example, Cross et al. used the EGFR-mutant lung cancer cell lines PC9, H1975, H1650, and H3255 to validate the function of EGFR inhibitor osimertinib, and their results indicated effective inhibition of EGFR phosphorylation [39]. Unfortunately, however, due to their extensive use and passaging from the time they were established, they do show differences in their genetic characterization of the parental tumor [40], thus limiting the reflection of the clinical response in drug screenings in a clinical trial setting.

### 2.2. Primary Cell Cultures

Due to the concern for potential loss or alteration in some of the genomic characters of the immortalized human lung cancer cell lines during passages, patient-derived primary cell cultures have been extensively studied during the past decade. These patient-derived primary cell cultures better preserve the tumor genomic profiles due to the relative recency of their establishment [65]. Moreover, primary cell cultures have a great advantage over immortalized cell lines for understanding the cellular interaction between various cell types including epithelial cells, fibroblast cells and immune cells. For example, Beverly et al. were able to establish an in vitro tumor microenvironment (TME) system using primary cell culture that allowed them to rapidly expand stromal progenitors from resected patient lung tumor specimens and showed that these progenitor populations retained the expression of pluripotency markers and secreted factors that are associated with cancer progression and tumor growth [66]. This model provides a valid tool to understanding the biological functions of these progenitor cell populations and develop strategies to inhibit the tumor growth and metastasis.

Another advantage to primary cell cultures is in the aspect of drug discovery and toxicity studies. Evaluating the efficacy and/or toxicity of candidate drugs or their therapeutic potential using primary cell cultures may provide a reflection closer to the predictive biological response in patients with identical or closely related disease conditions. As such, Benet et al. showed that using patient-derived primary cell culture combined with an immunofluorescence-based functional assay can effectively quantify tumor cell response to targeted therapy in mixed cell cultures [67]. An interesting study was conducted by Dandachi and her colleagues, where they obtained a primary cell culture from a patient with primary resistant LUAD. They identified two biologically profoundly distinct adenocarcinoma cell subpopulations. One they cultured in a spheroid culture system, while the other was only able to grow and proliferate under adherent conditions. They characterized these subpopulations and learned that the subpopulation from the spheroid culture system was strongly associated with the epithelial phenotype and expressed many of the cancer stem cell (CSC) markers, such as ALDH1 and CD133, while the adherent subpopulation was of a mesenchymal feature and did not express CSC markers. When testing this subpopulation with cisplatin, they demonstrated that the subpopulation from the spheroid culture was significantly more resistant to cisplatin compared to the adherent subpopulation. This study indicated that the primary LUAD cell culture was derived from a patient with resistant disease with an epithelial aggressive subpopulation of cells associated with stem cell features and resistance to therapy [68]. In summary, primary cell cultures can be invaluable because they more closely resemble the properties of cells in tissue and TME as well as allow analysis of cells from specific patient populations.

One frequent challenge when working with primary cell cultures is the lack of indefinite proliferative capacity, which can limit their utility in long-term culture of these cells. However, recent advances in cell culture techniques and the development of specialized culture media, scaffolds, and growth factors have made it possible to extend the lifespan of primary cells. As such, Odintsov and his colleagues established two cell line models from NRG1-rearranged (Neuregulin-1) lung adenocarcinoma samples and showed via transcriptome analysis the activation of the mTOR pathway in these NRG1 fusion-bearing samples. These cell lines were also more sensitive to mTOR pathway inhibitors (such as rapamycin), indicating potential therapeutic significance for patients for whom ERBB-directed therapy fails [69]. Studies indicate the potential advantage of primary cell cultures over established human lung cancer cell lines; however, the success rate in establishing patient-derived primary cell cultures is relatively low [67,70].

### 2.3. Conditionally Reprogrammed Cell

Conditional reprogramming (CR) is a type of cell culture that involves culturing the cells from patient samples with irradiated mouse cells in the presence of a Rho-kinase inhibitor (ROCK). The most common Rho kinase inhibitor is Y27632, and the most common irradiated mouse cell line used is Swiss 3 T3-J2 mouse fibroblasts [71]. Cells can quickly develop a stem-like character with high proliferation potential, retaining the original karyotypes under this specific condition [72]. This type of culture system (CR) is widely used for cells of epithelial origin.

One of the key advantages of the CR system is that it allows researchers (both basic and preclinical researchers) to describe the biological characteristics of cancer and explore and identify the relevant mechanism for drug resistance, maintenance, and establishment of tumors. It faithfully mirrors primary cancerous cells and can delineate the cellular, molecular, and genetic characterizations of cancer malignancy. As such, the CR system was used to establish primary cultures from NSCLC, and drug-response profiling was used to identify the histopathological subtypes of NSCLC-selective signal plasticity and associated therapeutic weaknesses [73]. For example, Brodovsky et al. established human ovarian and lung patient-derived xenograft (PDX) tumors into conditionally reprogrammed cell lines (CR-PDX). When comparing the genetic profile and histology of this CR-PDX to the parental tumors, they showed that CR-PDX maintained the original characteristics of the parental tumors with high fidelity [74]. Several studies used CR cells to elucidate the molecular players that have been shown to be crucial in cancer malignancy and thus establish a better target to abolish tumor growth. For example, Beglyarova et al. showed that MYC-ERCC3 interaction is a critical survival signal for pancreatic ductal adenocarcinoma (PDAC) via the use of conditionally reprogrammed patient-derived pancreatic cancerous cells [73]. Likewise, Yuan et al. used the CR system to identify a unique mutation from a duplication of the promoter and oncogene regions in the HPV-11 viral genome that were responsible for aggressive clinical features [75]. Alamri et al. used the CR technology to identify a novel therapeutic target, a gene fusion called KRT14-KRT5, in mucoepidermoid carcinoma and other salivary-gland neoplasms [76]. One advantage of using these CR cell lines is that besides retaining the genetic characteristics of the parental tumors, they are amenable to gene manipulation and drug screening and can even be used for implantation into immunodeficient mouse models for in vivo experiments [77]. One of the most common ways to obtain tumors is via biopsy; however, it is an invasive method, and thus, there are more and more studies focusing on obtaining tumor cells in a non-invasive manner, such as collecting cancer cells from other body fluids such as blood or urine. Jiang et al. reported an overall successful rate of 83.3% in establishing CRC of bladder cancer from urine [78].

### 2.4. Cancer Spheroids and Organoids

While the establishment of lung cancer cell lines over the past few decades has allowed the advancement of lung cancer research, it has also brought many potential challenges. As such, while cell lines with distinctive EGFR (epidermal growth factor receptor) mutations show different sensitivity to TKIs (tyrosine kinase inhibitors) [79], unfortunately, they fail in clinical trials using experimental therapies [65]. The introduction of immunocompromised mouse colonies in research has allowed the establishment of PDX mouse models that allow not only the maintenance of patient-derived cancer cells that do not survive in vitro for prolonged periods of time but also the establishment of the complex microenvironment a tumor would typically grow in, including a nutrient- and oxygen-rich blood supply capable of removing toxins, the extracellular matrix, the presence of other cell types, and growth factors. It allows tumors to promote blood vessel formation and to metastasize—something that cannot yet be achieved in in vitro settings.

Unfortunately, the inherent variability from mouse to mouse in establishing tumors leads to difficulty in establishing a standardized protocol. Furthermore, the lack of competent immune system in the PDX models indicates that they may not accurately reflect disease progression and the therapeutic response observed in immune-competent humans. For example, studies focusing on ICI-based therapies or other immunotherapeutic cancer vaccine research using these PDX mouse models will not be feasible to assess the efficacy of these drug studies.

Patient-derived lung cancer spheroids were first established by Eramo et al. in 2008 [80]. They were able to create a personalized 3D model to use to generate xenografts that recapitulated the histology of the parental tumors [80]. From cell composition to tissue structure, they are similar to human tumors; organoids can self-renew and self-organization, which can simulate the microenvironment of human cancer [81]. With the establishment of spheroids, many laboratories have focused on using patient-derived lung cancer spheroids for both in vitro and in vivo studies for drug testing and various molecular analyses [82,83,84,85,86]. The attempt by Endo et al. to generate NSCLC spheroids from tumor tissues or pleural effusion using Matrigel marked a step closer to optimizing organoid culture conditions [87]. These advancements further stimulated researchers to attempt to generate lung 3D cultures in a semi-structured environment using human respiratory epithelial cells isolated from nasal polyps. This method showed the generation of tubular structures containing cuboidal-shaped polarized cells, ciliated cells, secretory cells, and undifferentiated cells with epithelial cells that can contract [88]. However, using lung cancer spheroids for patient prognosis and for the ability to be able to select the right treatment of choice confidently is limited.

These studies prepared the road for lung organoid development. Following the path, various laboratories successfully established the first generation of normal lung organoids using human induced pluripotent stem cells (iPSC) [89,90,91]. Soon after, organoids were derived from adult lung primary cells and embryonic lung epithelial cells [92,93]. All these further enhanced our understanding of the culturing conditions with many trials. Clevers et al. were the pioneers in establishing lung cancer organoids (LCOs). The unique characteristics of these LCOs were that they retained the histopathology and the mutation profile of the original tumors and were amenable for both small-scale drug-screening experiments and orthotopic transplantation [94]. One reoccurring problem and disadvantage observed during these advancements is the competitive growth of normal epithelial cells. One possibility of overcoming this phenomenon is using media that lack the necessary factors to grow normal lung organoids [95]. Other possibilities include hand-picking tumor organoids [96,97] and deriving cancer cells from extrapulmonary sources such as metastases [94,98,99,100], pleural effusions, or PDXs [99,100,101,102]. While the generation of LCOs from lung cancer metastases has been shown to have a higher rate of success than from intrapulmonary samples [96], biopsy samples tend to be very small in size. Malignant effusion samples have shown promise as a source for LCOs, but the number of studies is still very low [99,100,101]. However, Kim et al. in their study showed that the LCOs they generated from effusion samples did, indeed, reproduce the genetic features of advanced LUAD and that they were amenable for predictive studies [99].

The unique characteristics mentioned above of the tumor organoids allow both basic and translational research applications. Several studies in various cancer types have used organoids to study critical steps in cancer initiation and progression, such as self-renewal, drug resistance, heterogeneity oncogenic transformation, circulating tumor cells [103,104,105,106,107,108], and drug screening [109]. The first study using lung organoids to obtain transcriptional and proteomic profiles or normal epithelial progenitors as compared to early-stage lung cancer was performed by Dost et al., where they used organoids that were derived from human iPSC and murine lung epithelial cells to model LUAD development [110]. Their study provided a comprehensive molecular landscape of K-RAS-driven lung tumorigenesis and, furthermore, encouraged other researchers to carry out studies on LCOs to provide new insights on the role of specific genes in lung cancer and/or on the biological functions of lung cancer cells [102,111,112,113].

The establishment of the LCO technology allowed researchers to perform the first drug-testing experiments within the context of various genomic alterations [114]. Sachs et al. were among the first to show that LCOs are, indeed, amenable to drug screening while being able to retain driver mutations of parental tumors with differential responses to chemotherapeutics and targeted inhibitors such as erlotinib, gefitinib, crizotinib, and alpesilib [94]. Kim et al. were able to establish a biobank that contained 80 LCOs and recapitulated the histology and genetic features of major lung cancer subtypes and responded to targeted drugs respective to their genomic alterations [99]. Various laboratories followed the establishment of biobanks composed of LCOs and tested both monoclonal antibody-based as well as antibody–drug conjugates (ADCs), ICIs, and targeted inhibitors and have found that LCOs are amenable for drug screening and that their responses correlated with their genetic profile, which represents the parental tumors’ genetic profile with high fidelity [109,115,116]. Interestingly, Li Z. et al. showed that several drugs were effective on LCOs without the related mutation. While this clearly indicates that not 100% of organoids preserve the genetic profile, drug testing using LCOs correlated with the predictive drug response as shown previously by PDO-based clinical trials [117,118]. It also indicates that routine drug testing on LCOs may disclose patients that could unexpectedly benefit from certain targeted treatments that otherwise their genetic profiling would not indicate [119]. Single-cell RNA (scRNA) sequencing and quantitative imaging technology have been used to assess the spatial, genomic, and transcriptome analysis of organoids at the single-cell level, which holds tremendous value in precision oncology in lung cancer. Studying the heterogeneity and evolution of tumor organoids via scRNA sequencing indicated that the CD44-positive subpopulation is responsible for drug resistance by hyper-activating the Jak-STAT signaling pathway in a hepatobiliary tumor [118]. Interestingly, in another study, Wang et. showed that the organoids from lung cancer patients’ malignant serous effusion match both the pathological characteristics and genomic profiling with the original malignant serous effusion with only a few somatic alterations. In vitro drug screening on these organoids was tailored to individual patients and was shown to discriminate between clinically sensitive and resistant patients [120].

### 2.5. Co-Culture System of Patient-Derived Immune Cells and Patient-Derived Tumors

Recent studies have shown the importance of various cellular and molecular components in the TME in prognosis and treatment responses to immunotherapy. Co-culture of LCOs with the systemic adding of key components of these factors allows researchers to define its contributions using in vitro models. Monoclonal antibody-based therapies that target the inhibitory receptors expressed by immune cells (i.e., ICIs) have shown remarkable response rates in various solid tumor malignancies [121]. Developing preclinical models that allow one to investigate the TME and guide clinical precision therapy is gaining more importance [122]. ICIs have emerged as a groundbreaking advancement in treating various cancers. To name a few, antiprogrammed cell death protein 1 (PD-1) and anticytotoxic T-lymphocyte-associated protein 4 (CTLA-4) have shown significant clinical efficacy in specific patient populations [123,124]. Many clinical trials indicate these therapies’ effectiveness [125] and, more importantly, have demonstrated exceptionally high efficacy in tumor types characterized by high mutational burden [126]. However, unfortunately, while these therapies focusing on ICIs show promise, their effectiveness, especially in solid tumors, is still very limited [127].

Studies have focused on establishing a co-culture system with peripheral blood mononuclear cells (PBMCs) and cancer cells. For example, Saraiva et al. could co-culture PBMCs and spheroids derived from the breast cancer cell line MDA-MB-230. They observed that the patient’s immune cells exhibited a wide range of antitumor responses and that this can be manipulated to improve their ability to lower the viability of tumor cells [128]. Co-culture of autologous tumor organoids and PBMCs is a great way to enrich tumor-reactive T cells from the PBMCs of patients. It can be used to assess the efficiency of killing matched tumor organoids. Dijkstra et al. showed in their study that co-culturing the autologous tumor organoids with PBMCs indeed can provide an unbiased strategy for the isolation of tumor-reactive T cells and can provide a means by which one can assess the sensitivity of tumor cells to T cell-mediated attack at the level of the individual patient [129]. Ma et al. used cell pellets from pleural effusions from patients that were characterized as oncogene-driven, based on if patients whose tumors had at least one driver oncogene (EGFR, MET exon 14 skip, or ErbB2 mutation, ALK, ROS1, or RET fusions), or non-oncogene-driven; these were used to co-culture with their respective autologous PBMCs to assess the blood biomarker profiles with respect to the effect of tyrosine kinase inhibitors (TKIs) and the responsiveness to ICIs [130]. The study revealed that ICI treatment activated additional immune cell types and that TKI treatment could either antagonize or enhance the effect of ICIs [130]. Table 2 summarizes the advantages, disadvantages, and key applications of these in vitro models.

## 3. In Vivo Models

The expansion of various in vitro models in cancer research has enabled us to advance in understanding basic tumor biology from initiation to metastasis and allowed the identification of the major molecular players that have an essential role in tumorigenesis. The sequencing of larger fragments ultimately allowed us to sequence the human genome along with the genomes of many other organisms and learn the genomic similarities between humans and mice, for example. These further encouraged researchers to establish mouse models for cancer research to understand tumor growth. All these led to the development of various mouse models from immunocompetent humanized animals that bear human tumors with a responsive immune system to learn the immune response and tumor interaction to immunocompromised animals that can tolerate cross-species cancer cells such as human cancer cells and nude mice strains that not only lack immune system but also are hairless, allowing one to be able to perform whole body imaging (Figure 2). This next section briefly summarizes the various mouse models that are frequently used and their potential in cancer research. Table 3 summarizes the advantages, disadvantages and key applications of these in vivo models.

### 3.1. Carcinogen-Induced Mouse Model

These models refer to particular cancer types developed in the animals via exposure to certain environmental risk factors, for example, carcinogenic chemicals, radiation, viruses of microbial flora, or physical stimuli [131]. One of the most significant advantages to these models is that they can simulate the tumor progression from the early stage, thus leading to the identification of general mechanisms of cellular alterations required for tumor formation. Additionally, they may be able to provide insights into tissue-specific features. In lung cancer studies, 4-(methylnitrosamino)-1-(3-pyridyl)-1-butanone (NNK)-induced mouse models are a great source to study lung cancer development due to cigarette smoking. Other compounds such as polyaromatic hydrocarbons as benzo(a)pyrene (BaP), N-nitroso-tris-chloroethylurea (NTCU), diethylnitrosamine, and 2-methycholanthrene (MCA) are also widely used for establishing chemically induced lung cancer mouse models [132]. For example, Wang et al. used NTCU on eight different strains of mice via skin painting and found that in five strains (SWR/J, NIH Swiss, A/J, Balb/cJ, and FVB/J), this method allowed for the establishment of lung small cell carcinoma (SCC). However, this method failed to establish tumor models in the inbred strains AKR/J, 129/svJ, and C57BL/6J [133]. Another study focused on repeated intratracheal injections of MCA in BC3Fl and DBA/2 mice strains and were able to induce squamous cell lung carcinoma [134]. When orally administering BaP to Swiss albino mice at 50 mg/kg dosage, twice a week for four weeks, Rajendran et al. were able to induce lung tumor models [135]. A similar study also succeeded where the researchers administered BaP at 100 mg/kg intraperitoneally into A/J mice to assess the chemoprevention efficacy of deguelin and silibinin [136]. Taken together, many of the compounds above are a great source when establishing lung cancer mouse models to study the effects of cigarette smoking, tumor initiation and progression, and the potential effects of currently available chemotherapeutics to combat lung cancer.

### 3.2. Syngeneic Mouse Models

Syngeneic mouse models are powerful resources that allow the generation of tumors in immunocompetent mice. Their ability to model the genetics of human disease is limited, but gene editing strategies can be used to engineer clinically relevant mutations. Notably, most human tumors with driver gene(s) mutation also tend to be coupled with genomic rearrangements, which these models still fail to recapitulate. Despite these limitations, these models are still very powerful, as one can generate orthotopic tumors, and these tumors will have similar histological features as their human counterparts, thus allowing one to study both the evolution of a complex TME and the assessment of various therapeutics [137,138]. Furthermore, assessing the response to various immunotherapies places these mouse models at the forefront of other immunocompromised mouse models. It has allowed researchers to understand the resistance mechanism of certain immunotherapies, such as anti-PD-1 [139]. Several other studies highlight the importance of these models, from assessing combination therapy with checkpoint inhibitors to identifying genetic players that correlate with pathways that, if targeted in combination, can reduce tumor growth and prolong the overall survival of patients. For example, Ajona et al. established syngeneic mouse models for lung cancer and showed that inhibition of the C5a/C5aR1 and PD-1 signaling have synergistic antitumor effects [140]. Meraz and his colleagues delivered TUSC2 (tumor suppressor that encodes a multikinase inhibitor and has been shown to be lost in non-small cell lung carcinoma) systemically by nanovesicles, which was shown to mediate tumor regression. Since TUSC2 is known to regulate immune cells, Meraz and his team assessed the TUSC2 efficacy on antitumor immune response alone and in combination with anti-PD-1 in K-RAS mutant syngeneic mouse models. Their results indicate that while TUSC2 alone significantly reduced tumor growth and prolonged survival compared with anti-PD-1 only, when combined, this effect was significantly enhanced and correlated with an increase in circulating natural killer (NK) cells and CD8+ T cells and a decrease in regulatory T cells (Tregs), myeloid-derived suppressor cells (MDSCs), and T cell checkpoint receptors PD-1, CTLA-4, and TIM-3 [141]. In summary, while it is not fully understood yet if syngeneic models truly reflect the human immune response, it is clear that these models are extremely useful in many aspects of cancer biology and allow researchers to obtain a plethora of functional assessments to pave the road for the development of novel therapeutics.

### 3.3. Transgenic/Genetically Engineered Mouse Models

The genetically engineered mouse model (GEMM) for cancers refers to an animal strain with manipulated genomic alterations, specifically the overexpression of an oncogene or the loss of a tumor suppressor gene function [142]. The GEMMs can be divided into transgenic and endogenous ones [143], allowing the investigation of the function of certain genes or pathways during tumor development or progression [144]. The ability to manipulate certain gene(s) in these models allow for further study to evaluate anticancer drug efficacy [145]. Furthermore, since these models are immunocompetent, they can also be invaluable for immune therapy assessments. For example, the K-RAS-LSL-G12D mouse model is an excellent source in lung cancer research as most lung cancer patients with a history of cigarette smoking are also known to have a mutation in the K-RAS gene. There are various ways to establish GEMs, most relying on engineered nucleases. These engineered nucleases are composed of sequence-specific DNA-binding domains fused to a non-specific DNA cleavage molecule [146,147]. The advantage of these chimeric nucleases is that they are able to execute genetic modifications in an efficient and precise manner by inducing targeted DNA double-strand breaks (DSBs) which then triggers the cellular DNA repair mechanism either via error-prone non-homologous end joining (NHEJ) or via homology-directed repair (HDR) to “repair” the cleavage [148]. Zinc finger nucleases (ZFNs) and transcription activator-like effector nucleases (TALENs) were at the forefront where the DNA-binding domain was fused with the cleavage domain of the FokI endonuclease [149,150]. Briefly, a pair of these nucleases bind to opposite strands of adjacent sequences separated by a short spacer sequence where the target site is located [149,150]. The usage of these nucleases is still limited because their construction requires modular assembly technology for generating the DNA-binding domains. The next most effective and simplest engineered nucleases are the CRISPR/Cas9 system. This system contains the Cas9 nuclease and so-called single guide RNA (sgRNA) M [150,151]. The single guide RNA is usually about 20 nucleotides in length and is complementary to the target site, followed by a tri-nucleotide protospacer adjacent motif (PAM) in the genome. This allows the Cas9 nuclease to be recruited to the target sequence [149,151]. Because of its simplicity and precision, CRISPR/Cas9 is currently the most prominent tool for genome engineering. Rakhit et al. used a Cre-regulated genetically engineered mouse model for lung adenocarcinoma development driven by K-RAS G12D (K-RAS-LSL-G12D mouse model). They tracked the release of cell-free DNA vs. cell-tumor DNA (cfDNA/ctDNA) and compared this with the tumor burden that was captured by micro-computed tomography (CT). To monitor ctDNA, they developed a droplet digital PCR (ddPCR) assay so that they can discriminate the K-RAS-lox-GD12 allele from the K-RAS-LSL-G12D and the K-RAS WT (wild type) alleles. They showed that the micro-CT correlated with endpoint histology and detected pre-malignant tumors with a combined tumor volume of 7 mm^3^ or larger [152]. By this model, they validated that cfDNA/ctDNA levels can be used as an early detection method for lung cancer, which correlated with mouse micro-CT measurement results in a consecutive test.

In another study focusing on the initiation and maintenance of lung cancer, researchers established two doxycycline-inducible transgenic mouse models: one bearing a point mutation in the EGFR gene (substitution of arginine for leucine) at exon 21 (EGFR L858R), while the other expressing a deletion of exon 19 (EGFR DeltaL747-S752). Both are common mutations in most lung cancer patients. When induced with doxycycline, they could express these EGFR mutants that lead to the development of lung adenocarcinomas. Interestingly, two weeks after doxycycline indication, the EGFR L8585R mutants showed diffuse lung cancer that resembled human bronchioalveolar carcinoma in contrast to the mutant-deletion mice (EGFR DeltaL747-S752) that developed multifocal tumors embedded in normal lung parenchyma. When they withdrew doxycycline or treated with first-generation EGFR TKI erlotinib, they observed tumor regression, indicating that these mutations are indeed required for tumor maintenance [153]. It is worthwhile to note that secondary mutations arising in tumors after an antitumor treatment is a very common phenomenon, and Politi et al. showed that GEMMs can simulate this. This was demonstrated by using the point mutant mouse model (EGFR L858R) that started treatment with erlotinib. In the study, they showed that initially, tumors were sensitive to erlotinib, but after multiple cycles of drug treatment, they became resistant. They correlated this resistance to a secondary mutation arising in the EGFR T790M and MET amplification [154]. Using an EGFR L858R/T790M transgenic mouse model, K-K Wong’s team showed that second-generation EGFR TKI afatinib (BIBW2992) induced tumor regression in xenograft and transgenic lung cancer models [155]. MET amplification is a resistance mechanism common to third-generation EGFR TKI osimertinib. Maraver’s lab recapitulated this acquired molecular resistance mechanism by generating an EGFR/MET transgenic mouse model and showed the addition of MET inhibitors to osimertinib-induced tumor regression [156]. These important preclinical data support the clinical testing of these drugs that have gained FDA approvals.

### 3.4. Cell Line-Derived Xenografts (CDXs)

CDXs are widely used for preclinical drug efficacy tests, pharmacokinetic and pharmacodynamic (PK/PD) correlation, and combination therapy. Human tumor cell line xenografts establish tumors when implanted in immunodeficient mice. In this model, it is feasible to investigate the efficacy of novel agents in inhibiting tumor growth and development and the mechanisms of action. Although implanting human tumors directly into mice is gradually becoming popular in antitumor drug discovery due to their accurate recapitulation of patient tumor features, CDX models are still proven to be an efficient method for in vivo study in terms of their excellent repeatability. There are approximately 300 cell lines for human lung cancer, most of which belong to NSCLC, and only a few belong to SCLC. The use of CDXs is important in understanding drug effect with respect to tumor inhibition. For example, one research group established CDXs by suspending 2 × 10^6^ A549 cells in 100 μL serum-free medium and inoculating subcutaneously into 4–5-week-old male BALB/C nude mice. It took about two weeks to reach a tumor volume of 100–120 mm^3^. Then they treated these mice with FEN1 inhibitor and cisplatin, respectively, or in combination. FEN1 is a major component of the base excision repair pathway for DNA repair systems and is important in maintaining genomic stability through DNA replication and repair. They showed that FEN1 is critical for the rapid proliferation of lung cancer cells, as when FEN1 is suppressed, a decrease in DNA replication and the accumulation of DNA damage were observed, leading to apoptosis.

How FEN1 was altered also dictated how the lung cancer cells responded to chemotherapeutic drugs. If they targeted FEN1 with a small-molecule inhibitor, they noticed the enhanced therapeutic effect of cisplatin [157]. In another study, Kim et al. used CDX models to identify the Hedgehog pathway transcription factor GLI1 as a critical driver for lung squamous cell carcinoma. Human lung cancer datasets indicated that GLI1 mRNA is highly expressed and correlates with poor prognosis. While inhibitors targeting the Hedgehog pathway did not seem to alter the expression of GLI1, interestingly, modulation of the PI3K/Akt axis allowed them to modulate GLI1 expression. When these tumors were grown in CDXs, they observed that tumor growth could be attenuated in the tumors that harbor amplification of PIK3CA (PI3K gene) by antagonizing GLI1 and PI3K, further supporting their regulatory function [158]. It is clear from these and other similar studies that CDXs are invaluable in assessing drug effects and can aid in identifying molecular players that may have a crucial role in cancer initiation and progression. One drawback of CDXs is that they rely on established human cell lines, which are mainly of Caucasian origin. Thus, our research is gaining an understanding of tumor characteristics and drug responses, which are narrowed down only to Caucasian populations, thus limiting the predictable response in other ethical groups. Furthermore, the tumors formed in CDXs lack the TME that would occur in a real setting.

### 3.5. Patient-Derived Xenografts (PDXs)

The introduction of immunocompromised mouse colonies in research has allowed the establishment of PDX mouse models that allowed both the maintenance of patient-derived cancer cells that do not survive in vitro for prolonged periods and also the establishment of the complex microenvironment a tumor would normally grow in, including a nutrient- and oxygen-rich blood supply capable of removing toxins, the extracellular matrix, the presence of other cell types, and growth factors. The PDX model inoculates a patient’s tumor tissue directly into immunodeficient mice by subcutaneous or orthotopic transplantation. The most commonly used strain for the establishment of PDX models is the NOD-scid IL2Rgammanull (NSG) mice that lack functional B and T cells and natural killer cell activity and are the most immunodeficient yet are physiologically durable and can establish consistent engraftment of human primary tumors; they are invaluable for in vivo drug testing [159]. This method retains tumor tissue heterogeneity because it is not artificially cultured; tumors’ biological characteristics remain more complete, so these models can better simulate the reality of tumor patients [160,161]. Several genomic databases provide abundant resources for PDX gene characterization (Table 4).

As PDXs have been increasingly used in precision oncology for lung cancer patients, it is crucial to standardize the development and genomic analysis tool to characterize PDX tumors and compare to donor patient tumors. Meehan TF et al. summarized a criterion to evaluate PDXs [165]. A PDX has three main applications. The first is basic research. Some scholars have studied the gene expression differences between successful and unsuccessful tumor tissue modeling and found that there are 163 abnormally expressed genes, and these genes are mainly concentrated in signal pathways such as cell cycle/mitosis and cell proliferation [166]. Several genomic data analysis workflows and guidelines have been reported as well. Genomic data analysis can help researchers discover key oncogenic mutation and new tumor biomarkers [167]. The second application is preclinical. Compared with the CDX model, the clinical relevance of the PDX model is up to 89–90% as PDXs can better simulate human responses to drugs and indicate a more reliable way to assess preclinical drug analysis. Furthermore, PDX models can evaluate molecular targeted therapy [168], chemotherapy [169,170], and ADCs [171]. Using PDXs, researchers found that EGFR-activating mutation PDX models were very sensitive to gefitinib, while KRAS mutation PDX models were not [172]. Our team recently utilized several PDX models to examine the effects of combined TKI and statins in TKI-resistant patients. Simvastatin demonstrated a powerful antitumor effect in tested LUAD cell lines and PDX tumors, irrespective of tumor genotypes [173]. Nitin Roper et al. found a osimertinib and savolitinib combination works best for osimertinib-resistant EGFR-mutant tumors with MET pathway activation in PDX models [174]. The third application is in the clinical phase, where cancer patients often develop drug resistance, leading to treatment failure. Developing an efficient method to test drug efficacy in a single dosage or combination is urgent. Since the PDX can largely retain the tumor biological characteristics in patients and is a reliable model when it comes to testing drug efficacy, it could be used to guide clinical treatment regimes. For example, a study reported that the PDX models of subrenal capsule xenografts showed consistent responses to the chemotherapy when correlating it with their clinical data [170]. Compared with other existing models, PDXs could preserve TME to a great extent; however, compared with the high success rate of CDX, the PDX success rate is significantly different in different tumors. The highest success rate when establishing a PDX is of acute lymphocytic leukemia, which is up to 64%. In contrast, the success rate of lung cancer PDX is lower [175]. In addition, the establishment of the PDX model is also relatively long. The first generation of tumors grown from tumor tissues taken from patients (P0 generation) and implanted in immunodeficient mice takes about 1–8 months to establish [176].

Many factors affect the success of a PDX. The most important factors include the quality and viability of tumor tissue [177]. Others include the degree of immune deficiency in transplanted mice, the interval between patients’ adjuvant chemotherapy, the implantation time of tissues, the method of tumor extraction, and the stage and type of tumor. As such, it was reported that the success rate of the PDX model of lung squamous cell carcinoma tissue is higher than that of lung adenocarcinoma [178]. The malignancy of the tumor tissue has a more significant effect on the modeling success rate. The higher the degree of malignancy, the higher the modeling success rate [179]. It is important to note that the method used to establish patient-derived xenografts (PDXs) from tumor tissue sections can impact the success rate of engraftment. Specifically, the tumor can be sectioned into smaller fragments, enzymatically digested, or physically manipulated into a single-cell suspension, each approach offering its own benefits and drawbacks. One advantage of using tumor fragments over a single-cell suspension is that fragments can retain cell–cell interactions and some tissue architecture of the original tumor, thereby better mimicking the TME. In contrast, the advantage of a single-cell suspension is that it allows scientists to collect an unbiased sample of the entire tumor rather than inadvertently selecting spatially enriched subclones during analysis or tumor passaging [180]. However, establishing a single-cell suspension requires tumors to be either enzymatically digested or mechanically chopped into fine pieces, which may cause the cells to undergo anoikis, thus affecting their viability and the success of engraftment. Another potential disadvantage of PDX models is that during tumor growth in the animal, mouse-derived stroma gradually replaces the human stroma after three to five passages, leading to a loss of the model’s original biological characteristics [181]. Additionally, PDX models require immunodeficient mice, which limits studies that could benefit from evaluating the effectiveness of immunotherapeutic agents.

### 3.6. Humanized Mouse Models

As immunotherapy becomes an increasingly important treatment paradigm, the clinical response to these therapies remains highly variable. We still lack a full understanding of their mechanisms of action and specific biomarkers of response. Consequently, there is an unmet need for in vivo models that can replicate the interactions between the human immune system and tumors. One of the major limitations of using the NSG mouse strain for PDX models is the absence of a competent immune system, meaning they may not accurately reflect disease progression or therapeutic responses that would occur in immune-competent humans. Over the past decade, significant progress has been made in developing preclinical models for evaluating cancer immunotherapies. Recently developed humanized mouse models provide a unique tool for assessing the antitumor response of the human immune system to checkpoint inhibitors [182]. Currently, there are two primary approaches for establishing humanized mouse models, both of which require sublethal irradiation before the transplantation of immune cells [182,183] (Table 5).

One of these methods uses human PBMCs (Hu-PBMCs) or tumor-infiltrating lymphocytes, while the other method requires human hematopoietic stem and progenitor cells (Hu-HPSCs) [184]. For the Hu-PBMC-derived model, first, one has to isolate PBMCs, which is usually performed using Ficoll—which is a hydrophilic polysaccharide that allows the plasma and lymphocytes, erythrocytes, and PBMCs to separate based on their densities following centrifugation. The isolated PBMCs are then transplanted into the previously irradiated mice via intravenous administration. The most significant advantage of this model is that it is relatively easy and takes about 4 weeks to establish. Studying T cell function in this model is of benefit as human CD3+ cells, including both CD4+ and CD8+ subsets, are the most abundant cell population that survives the engraftment process; however, FOXP3+CD25+CD127low regulatory T cells (Treg) are only detectable for the first 2–4 weeks post-injection before they become undetectable [185]. Human innate cell populations (such as myeloid and NK cells) survive for the first few days in the animals and become undetectable both in the circulation and tissues. Interestingly, CD19+ B cells are maintained at low levels in specific sites such as the sleep and bone marrow for several weeks. Moreover, human IgGs can be detected in the peripheral blood of these animals for their whole lifespan [186]. However, the drawback of this Hu-PBMC-derived model is that immune therapy studies can only be performed in a relatively short period due to the graft-versus-host disease that happens faster in this model than in the other models. Lin et al. established this type of mouse model in their study to determine the efficacy of PD-1/PD-L1 immunotherapies and have showed this PBMC-derived PDX model was an invaluable tool for their study [181]. They also established Hu-HPSC models by first reconstituting the human immune system by transferring human CD34+ hematopoietic stem and progenitor cells into the mice. Once established, they then transplanted human lung cancer cells [182]. While the establishment of the Hu-HPSC takes a longer time, it can also be used for long-term immune therapy surveillance studies. The advantage of this mouse model is that they have a more complete hematopoietic system that includes innate immune cells, adaptive immune cells, and even low numbers of red blood cells and platelets [187], and GVHD is uncommon. CD34+ HSPCs are most reliably obtained from umbilical cord blood (UCB) or peripheral cells mobilized in response to G-CSF (CSF3), fetal liver tissue, and bone marrow [188]. It is important to note that many parameters can affect the engraftment of CD34+ HSPCs into immunodeficient mice, for example, the genetic background of the strain, the age of the animal, the source of CD34+ HSPCs, the route of injection and the number of CD34+ HSPCs, and the regimen used [189,190,191,192,193]. In addition, studies indicate that the source of CD34+ HSPCs influences the functionality of human T cells developed in engrafted mice, for example, fetal CD34+ HSPCs give rise to T cells with greater immune tolerance versus those that received adult CD34+ HSPCs [194]. Several other limitations are important to consider. These include the incomplete development of mature human innate cell lineages (monocytes, macrophages, DCs, and NK cells), incomplete functionality of the human B cells, and the absence of human leukocyte antigen (HLA) expression essential for developing HLA-restricted T cells. Several efforts to improve humanized mouse models have focused on incorporating various primitive immune cells to better evaluate immune response dynamics, understand mechanisms of immune evasion, and assess both the efficacy and potential adverse effects of immunotherapies [195,196]. However, these models are limited by the availability of donor resources.

## 4. Conclusions

Despite substantial progress in developing various preclinical models for functional precision research, each model discussed herein has its own limitations. Figure 3 summarizes the clinical applications for key in vitro and in vivo lung cancer models described in this review. Traditional models often fail to capture the full spectrum of human immune responses, which can limit the predictability of clinical outcomes. The improvement of preclinical models that accurately translate human immunity remains a top priority in cancer research [197]. New models, such as humanized PDXs with reconstituted immune systems and LCO–immune cell co-cultures, have shown promise in bridging this gap. By better replicating human immunity, these models are crucial for the development of precision immunotherapies tailored to individual lung cancer patients, ultimately improving therapeutic efficacy and reducing adverse effects. 

## Figures and Tables

**Figure 1 cancers-17-00022-f001:**
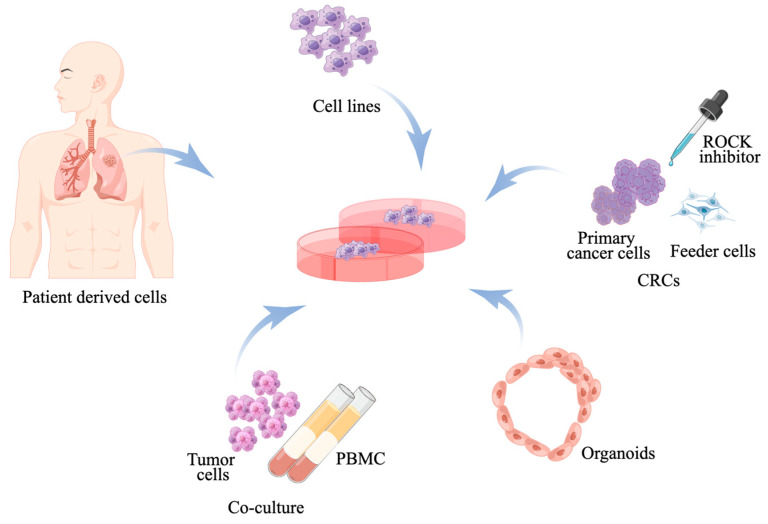
Schematic representation of the various in vitro methods. Immortalized cell lines, primary cell cultures, and organoids established from lung cancer patients serve as essential tools in precision oncology research. Each model offers unique insights into lung cancer biology and treatment responses. Co-culture of patient-derived lung tumor cells with PBMCs isolated from the same patient’s whole blood provides a more physiologically relevant model by incorporating the patient’s immune cells, allowing for real-time study of immune–tumor interactions (by Figdraw.com, accessed on 25 August 2022). Abbreviations: CRCs, conditional reprogramming cultures; PBMC, peripheral blood mononuclear cell; ROCK, Rho kinase.

**Figure 2 cancers-17-00022-f002:**
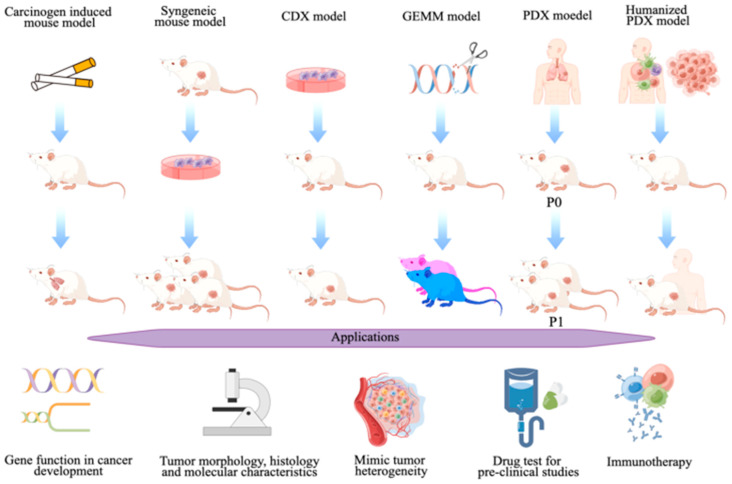
A schematic representation of the in vivo mouse models currently available for cancer research. Carcinogen-induced mouse models are induced to develop certain types of cancer by exposing them to certain environmental risk factors (carcinogenic chemicals, radiation, etc.). Syngeneic mouse models are immunocompetent animals that bear tumors of mouse origin. CDX models are immunodeficient mouse models, and tumors are implanted to assess drug function on a tumor. GEMM models are mouse strains that have been manipulated genetically either by the overexpression of an oncogene or by the loss of a tumor suppressor gene function. PDX models are immunocompromised animals implanted with tumors of human origin. Humanized PDX models can represent the human immune system to a certain extent along with tumors of human origin to study tumor–immune system interactions (by Figdraw.com; accessed on 27 August 2022).

**Figure 3 cancers-17-00022-f003:**
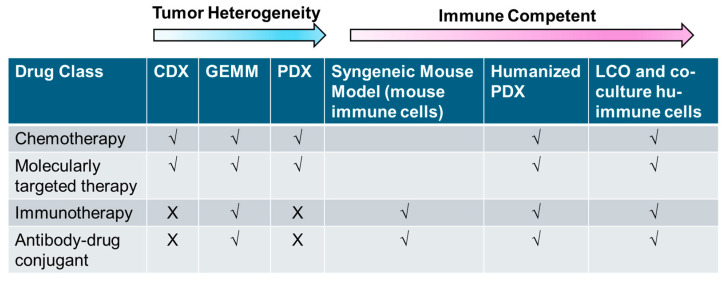
Summary of key preclinical models for precision lung cancer research.

**Table 1 cancers-17-00022-t001:** Summary of the most commonly used human lung cancer cell lines.

Cell Line	Histology	Driver Oncogene(s)	Other Mutations	Reference
H1975	Adenocarcinoma	EGFR L858R, T790M	PIK3CA, TP53	[41]
H3255	Adenocarcinoma	EGFR L858R	TP53	[42,43]
HCC2935	Adenocarcinoma	EGFR Exon19 del (E746-A750)	TP53, APC	[44]
HCC4006	Adenocarcinoma	EGFR Exon 19 del (L747-E749)	TP53, PIK3CA	[45]
H1650	Adenocarcinoma	EGFR Exon19 del (E746-A750), T790M	TP53	[43]
HCC827	Adenocarcinoma	EGFR Exon19 del (E746-A750)	TP53	[43,46]
PC9	Adenocarcinoma	EGFR Exon 19 del (E746-A750)	TP53	[47]
H1573	Adenocarcinoma	KRAS G12A, NRASQ61K	PTPN1, TP53	[48]
H23	Adenocarcinoma	KRAS G12C	TP53, ATM, STK11	[49]
H460	Large cell carcinoma	KRAS Q61H	STK11, PIK3CA, TP53	[50]
A549	Adenocarcinoma	KRAS G12S	STK11, TP53	[41]
H2122	Adenocarcinoma	EML4-ALK variant 3a/b	TP53	[51]
H358	Adenocarcinoma	KRAS G12C	CTNNB1, TP53 deletion	[49]
H1299	Adenocarcinoma	N-RAS Q61K	TP53 deletion	[49,52]
H596	Adenocarcinoma	MET exon 14 skipping	PIK3CA, RB1, TP53	[53]
H522	Adenocarcinoma	KRAS G12S	TP53	[50]
H2228	Adenocarcinoma	EML4-ALK fusion v3 ALK-PTPN3	TP53	[54]
H661	Large cell carcinoma	ARHGAP35 K179* mutation	CDKN2A, LASP1, TP53	[55]
H2126	Adenocarcinoma	-	SMARCA4, TP53	[56]
H1437	Adenocarcinoma	-	TP53	[56]
H1563	Adenocarcinoma	-	CDKN2A	[56]
H661	Large cell carcinoma	-	CDKN2A, LASP1, TP53	[56]
H1770	Carcinoma	-	TP53	[57]
H2170	Squamous cell carcinoma	-	RHOA, TP53	[58]
H69PR	SCLC	-	PIK3X, TP53, RB1	[59]
DMS235	SCLC	-		[60]
H2066	SCLC	-	TP53	[61]
COR-L279	SCLC	-	EP300, TP53	[62]
SHP-77	SCLC	-	ABL1, KRAS, RAC1, TP53	[63]
NCI-H727	SCLC	-	PKD1L-TNS3 fusion, KRAS, TP53	[64]

Note: Driver oncogenes are defined as EGFR: epidermal growth factor receptor; KRAS: Kirsten rat sarcoma; HER2: human epidermal growth factor 2; EML4-ALK: echinoderm microtubule-associated protein-like 4-Anaplastic lymphoma kinase; MET: mesenchymal epithelial transition; ROS: c-ROS oncogene 1; BRAF: v-raf murine sarcoma viral oncogene homolog B; NRAS: neuroblastoma RAS viral; SCLC: small cell lung carcinoma.

**Table 2 cancers-17-00022-t002:** Summary of in vitro models.

In Vitro Models	Advantages	Disadvantages	Applications
Human lung cancer cell line models	Inexpensive, scalable, and widely available.	Limited representation of parent tumor heterogeneity; no TME.	Study cancer at molecular and cellular levels.
Primary cell cultures	Closer to the patient genomic profile.	Low success rate in establishing patient-derived primary cell cultures.	Study cancer at molecular and cellular levels.
Conditionally reprogrammed cell	Remaining the original karyotypes.	No TME.	Study cancer at molecular and cellular levels.
Cancer spheroids and organoids	Highly mimic original histopathology of tumors, rapid and robust growth.	Expensive, lack of TME and tumor heterogeneity.	Study self-renewal, drug resistance, heterogeneity oncogenic transformation, and drug screening.
Co-culture system of patient-derived immune cells and patient-derived tumors	Mimic the TME.	Difficulty in reproducing results and interpreting results.	Study the interaction between immune system and tumor cell.

Abbreviations: TME, tumor microenvironment.

**Table 3 cancers-17-00022-t003:** Summary of in vivo models.

In Vivo Models		Advantages	Disadvantages	Applications
Carcinogen-induced mouse models		Tumor formation time is similar to human tumor growth progress.	1. Relatively long latency. 2. Uncontrollable experimental results. 3. Not popular anymore.	Study tumor morphology, histology, and molecular characteristics.
Xenografts	CDX	Readily available, over 300 models, cost-effective, easy to manipulate genetically, and widely used for high-throughput screening of drugs.	1. Limited representation of tumor heterogeneity and microenvironment; results may not always translate to clinical outcomes. 2. Except the subcutaneous implantation, other transplant methods are technically difficult and need special technology to monitor the tumor growth.	Best suited for initial drug screening and mechanistic studies involving genetic and molecular pathways.
	PDX	Retain the genetic, histological, and morphological features of original patient’s tumors.	1. Require immunocompromised mice, which is not suitable to evaluate immunotherapy. 2. Snapshot of patient tumor that cannot reflect the heterogeneous of whole tumor. 3. Orthotopic transplantation requires higher technical skill, has high cost, and needs in vivo imaging tool to monitor the tumor growth. 4. Takes 1–9 months to establish a model. 5. Mouse-derived cells gradually replaced human stromal cells.	Valuable for validating drug efficacy of chemotherapy and/or molecularly targeted therapy for individual patients and tailoring treatment plans for specific genetic profiles.
GEMM		Mimic genetic mutations observed in human cancers, allowing for the study of tumor initiation, progression, and therapy resistance within an immune-competent context.	1. Time-consuming and expensive to develop; can lack the full range of human tumor heterogeneity. 2. CRISPR/Cas9 system can evaluate off-target activity.	1. Study the function of tumor gene mutations and mechanisms of drug resistance. 2. Evaluate the mutation effect of immunotherapy.
Syngeneic models		1. Genetically identical to the host, enabling studies for immunotherapy and drug toxicity within a fully functional immune system. 2. Cost-effective compared to humanized models.	1. Tumor lines are typically murine in origin, which may not accurately reflect human tumor biology. 2. Adapt to mouse biology only, not sure if the outcome is suitable for the human immune system.	Useful for evaluating the interaction between tumor cells and immune cells, as well as testing immunotherapeutic agents.
Humanized PDX models		1. Hu-PBMC model is fast to grow. 2. Incorporate a functional human immune system, allowing for long-term research and low or miner GVHD rate.	1. Complex and expensive to establish. 2. Variability in immune reconstitution can affect reproducibility (low reconstitution rates of NK and B cells). 3. Short study period due to mouse developing GVHD.	Ideal for studying ICIs, CAR-T cell therapies, ADC, and other immunomodulatory treatments.
Patient-derived LCOs		1. Offer a high success rate in establishing cultures that maintain the histopathological and genomic fidelity of primary tumors. 2. Enable rapid drug screening and correlate well with clinical outcomes.	Organoids may not fully replicate the TME, including interactions with the immune system and stroma.	Emerging as a crucial tool for personalized medicine, facilitating the testing of various treatment regimens and helping guide clinical decision-making.

Abbreviations: CDX, cell line derived xenograft; PDX, patients-derived xenograft; SCLC, small cell lung cancer; NSCLC non-small cell lung cancer; GEMM, genetically engineered mouse model; PBMC, peripheral blood mononuclear cell; IV, intravenous; IP, intraperitoneal; IF, interfemoral; HPSC, hematopoietic stem and progenitor cells; GVHD, graft-versus-host disease; NK, natural killer, TME, tumor microenvironment.

**Table 4 cancers-17-00022-t004:** Resources for PDX and mouse models.

Database	Website	Information	Reference
Mouse Genome Database (MGD)	http://www.informatics.jax.org/	Gene characterization, nomenclature, mapping, gene homologies among vertebrates, sequence links, phenotypes, allelic variants and mutants, and strain data	[162]
Gene Expression Database (GXD)	http://www.informatics.jax.org/expression.shtml	Gene expression information from the laboratory mouse	[163]
Mouse Models of Human Cancer database (MMHCdb)	http://tumor.informatics.jax.org/mtbwi/index.do	Spontaneous and induced tumors in mice including GEMM, PDX	[164]

Abbreviations: GEMM: genetically engineered mouse model; PDX: patient-derived xenograft.

**Table 5 cancers-17-00022-t005:** Comparison of the different humanized mouse models.

Name	Method	Advantages	Disadvantages
Humanized PBMC (hu-PBMC) Mouse Models	Tail vein injection of hu-PBMCs, which include lymphocytes (T, B cells and NK cells), neutrophils, and monocytes	Easy and fast to build Cost-effective Idea model for studying T cell immunity which could be maintained by cytokines or GM-CSF or IFNγ Short term study model	Fast GVHD (3–9 weeks) B cells reconstitute rates are low
Humanized CD34+ (hu-HPSC) Mouse Models	Tail vein injection of hu-HPSCs, which include hematopoietic stem cells (HSC) and hematopoietic progenitor cells	Multi-lineage engraftment Robust T cell maturation Idea long-term study model without GVHD	Longer time to build More expensive NK and B cell reconstitute rates are low
Knock-in Humanized Mouse Models	Knock-in human gene to replace murine gene	Fully functional immune system	Long time to build High cost
Human Fetal Bone, Liver, and Thymic Tissue (BLT) Engraftment	Subcapsular injection of HLA-matched fetal thymus or other immune organs into kidney capsule; more intact TME	Best mimic the human TME for immunotherapeutics	Limited resources from fetal thymus Ethical report
Spleen Mononuclear Cell (SPMC) Engraftment	Intraperitoneal injection of single cells from donor splenic tissue; More B cells and TM cells than PBMCs	Prediction of CRS while minimizing GVHD Difficult to build	Limited resources for deceased spleen donor

Abbreviations: CRS, cytokine release syndrome; GM-CSF, granulocyte–macrophage colony-stimulating factor; GVHD, graft-versus-host disease; HLA, human leukocyte antigen; TM, memory T cells; TME, tumor microenvironment.

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
