# Peer review of "Preclinical Models for Functional Precision Lung Cancer Research"

_cancers, 2024, doi:10.3390/cancers17010022_

Round 1
Reviewer 1 Report
Comments and Suggestions for Authors
The review: "Preclinical Models for Functional Precision Lung Cancer Research" is overall well written; it provides a good picture of the state-of-the-art of preclinical model in lung cancer.
I would recommend for publication.
Minor comments:
Some small typos here and there:
-table 1 cell line H661 repeted twice,
-line 213 "with",
-table 2, syngenic model third column (have a look),
-line 316, 5 strains...this method,
-paragraph " Transgenic/Genetically engineered mouse models": when mention GEEM used to study mechanisms of resistance to TKIs ( Erlotinib) authors shouldmention EGFR L858R+T790M mouse model from K.K. Wong's Lab ( Oncogene. 2008 Apr 14;27(34):4702–4711. doi: 10.1038/onc.2008.109) as well as the EGFR-MET mouse model ( resistant to Osimertinib) from Maraver's Lab (Cancers 2021, 13(14), 3441; https://doi.org/10.3390/cancers13143441)
Author Response
We appreciate the enthusiastic review and have made the following minor revisions.
Reviewer 1: minor revisions.
Comment 1: -table 1 cell line H661 repeated twice;
Response 1: Thank you for pointing this out. We’ve deleted the duplicated H661 cell line.
Comment 2: -line 213 "with";
Response 2: Thank you for pointing this out. We have revised this sentence as below, now in line 351-353. “While this clearly indicates that not 100% of organoids preserve genetic profile, drug testing using LCOs correlated with the predictive drug response as shown previously by PDO-based clinical trials [118, 119].”
Comment 3: -table 2, syngeneic model third column (have a look)
Response 3: We have revised the sentence in the third column of the syngeneic model. Previous Table 2 became Table 3 after adding a new table summarizing the characteristics of in vitro models.
Comment 4: -line 316, 5 strains...this method;
Response 4: Thank you for pointing this out. We have put in a period to complete the sentence as below. Please see the update in line 456-458.
“For example, Wang et al. used NTCU on eight different strains of mice via skin painting and found that in five strains (SWR/J, NIH Swiss, A/J, Balb/cJ, and FVB/J). This method allowed for the establishment of lung small cell carcinoma (SCC).”
Comment 5: -paragraph " Transgenic/Genetically engineered mouse models": when mentioning GEEM used to study mechanisms of resistance to TKIs (Erlotinib) authors should mention EGFR L858R+T790M mouse model from K.K. Wong's Lab ( Oncogene. 2008 Apr 14;27(34):4702–4711. doi: 10.1038/onc.2008.109) as well as the EGFR-MET mouse model ( resistant to Osimertinib) from Maraver's Lab (Cancers 2021, 13(14), 3441; https://doi.org/10.3390/cancers13143441)
Response: Thank you for your suggestions. We added these two references in the paragraph " Transgenic/Genetically engineered mouse models." Please see updates in lines 563-570 as below.
“Using an EGFR L858R/T790M transgenic mouse model, K-K Wong’s team showed that second generation EGFR TKI afatinib (BIBW2992) induced tumor regression in xenograft and transgenic lung cancer models[156]. MET amplification is a common resistance mechanism to third generation EGFR TKI osimertinib. Maraver’s lab recapitulated this acquired molecular resistance mechanism by generating an EGFR/MET transgenic mouse model, and showed the addition of MET inhibitors to osimertinib induced tumor regression[157]. These important preclinical data support the clinical testing of these drugs that have gained FDA approvals.”
Reviewer 2 Report
Comments and Suggestions for Authors
The authors conducted a comprehensive review to provide an overview of various preclinical models used in functional precision lung cancer research. This study meticulously covers both in vitro and in vivo models, presenting highly informative and well-written content. In particular, the authors offer detailed insights into in vivo models, including comparisons between different models and their applicability to various research types involving different drug classes, such as chemotherapy, targeted therapy, immunotherapy, and ADCs (Figure 3). Below are several comments to enhance clarity and improve reader comprehension prior to publication:
The authors presented the aims of this study, which emphasize the advantages, limitations, and applications of preclinical models in advancing personalized medicine. However, while the application aspects are considered crucial, the in vitro section seems to lack sufficient detail. Although the primary focus of this study is on in vivo models, it would be beneficial to include more information on the applications of cell lines, cell cultures, and CR.
For cancer cell line models, it is worth noting that there are many SCLC models, including both adherent and floating cells, which are crucial for drug screening. Please consider adding some SCLC lines to Table 1.
The authors provided a detailed overview of lung cancer organoid models. It would be helpful to include a comparison between cell lines and organoids in their applications for preclinical studies, highlighting the differences.
Please expand on GEMM by including more details on the methodology and types of genomes editing used. This additional information will help readers gain a more comprehensive understanding.
Lastly, please provide more information on the transition from patient tissue to the creation of PDX models, specifically focusing on factors that influence the successful establishment of PDX models.
Author Response
Thank you for the insightful thorough reviews. Please see the responses below.
Comment 1: The authors presented the aims of this study, which emphasize the advantages, limitations, and applications of preclinical models in advancing personalized medicine. However, while the application aspects are considered crucial, the in vitro section seems to lack sufficient detail. Although the primary focus of this study is on in vivo models, it would be beneficial to include more information on the applications of cell lines, cell cultures, and CR.
Response: We agree and have revised our manuscript at these sections extensively: adding the details on Sections 2.1 (human cell lines), 2.2 (primary cell culture), 2.3 (conditionally reprogrammed cells) and 2.4 (cancer spheroids and organoids).
Comments 2: For cancer cell line models, it is worth noting that there are many SCLC models, including both adherent and floating cells, which are crucial for drug screening. Please consider adding some SCLC lines to Table 1.
Response 2: We have added a section on describing SCLC in Section 2.1, and added a few common SCLC lines in Table 1.
Comments 3: The authors provided a detailed overview of lung cancer organoid models. It would be helpful to include a comparison between cell lines and organoids in their applications for preclinical studies, highlighting the differences.
Response 3: We have revised our manuscript and included more information on cell lines and organoids. A new Table 2 has been added summarizing the advantages, disadvantages and key applications of these in vitro models.
Comments 4: Please expand on GEMM by including more details on the methodology and types of genomes editing used. This additional information will help readers gain a more comprehensive understanding.
Response 4: We agree and have revised the Section 3.3 for “Transgenic/Genetically engineered mouse models”, with adding the types of different genome editing technologies per references 147-152.
Comments 5: Lastly, please provide more information on the transition from patient tissue to the creation of PDX models, specifically focusing on factors that influence the successful establishment of PDX models.
Response 5: Although the manuscript already contains factors that influence the success for engraftment, we have included more and their advantage and disadvantage as a model in Section 3.5.
Reviewer 3 Report
Comments and Suggestions for Authors
In this manuscript, Yu et al. provide a summary of the various in vitro and in vivo models that are used to provide individualised cancer treatment for lung cancer. The authors highlight the advantages, limitations and applications of preclinical models and technological innovations to advance personalised medicine. Overall, the authors have addressed a very interesting research topic and the manuscript is clear and well written. The bibliographic collection on the topic is very recent and adequate for the purpose. The included figures and tables allow the reader a better understanding of the subject.
Although minor revisions are needed, I think the manuscript deserves to be accepted for publication. I wish the authors good luck in the publication process.
Minor revisions:
Line 119: NRG1 is reported just as abbreviation. Please, add the full name, Neuregulin-1 (NRG1).
Line 151: There is a space after the reference
Table 2: There is a superfluous 'an' in point 1 of the third column.
Line 301: Add between the abbreviations LCO
Line 507: Add spaces between the reference and the words
Author Response
Thank you for your enthusiastic review.
Comments 1: Although minor revisions are needed, I think the manuscript deserves to be accepted for publication. I wish the authors good luck in the publication process.
We appreciate the enthusiastic review and have made the following minor revisions.
1. Line 119: NRG1 is reported just as abbreviation. Please, add the full name, Neuregulin-1 (NRG1). Done
2. Line 151: There is a space after the reference
Thank you for pointing this out. Done
3. Table 2: There is a superfluous 'an' in point 1 of the third column.
This has been deleted.
4. Line 301: Add between the abbreviations LCO
Thank you for pointing this out. Done
5. Line 507: Add spaces between the reference and the words
Thank you for pointing this out. We have corrected the sentences now in line 671.
Reviewer 4 Report
Comments and Suggestions for Authors
I enjoyed the paper titled "Preclinical Models for Functional Precision Lung Cancer Research" by the respected team. They have done their best to address the precision medicine application in lung cancer. There are two major and important components which are missing from the current version:
1- It is very well know that "Metastasis" is the major cause of death by lung cancer. The respected authors are expected to address how they consider this important topic in their paper and different models.
2- Please consider how precision medicine can over come the chemotherapy and radiotherapy resistance in lung cancer or how it can help to improve the patient response.
3- The current topic could improve the overall outcome of research approach in lung cancer and also could be first steps to decrease the mortality rate in this specific type of cancer. How the author see the precision medicine and personalized medicine will benefit the patients for this approach.
4- How does the application of precision medicine will improve the health policy in future in this field?
5- Please provide a table compare advantage and disadvantage of each model they discussed in the paper.
Author Response
Thank you for your insightful review.
Comment 1: - It is very well known that "Metastasis" is the major cause of death by lung cancer. The respected authors must address how they consider this important topic in their paper and different models.
Response: We truly appreciate your thoughtful feedback! It’s great to see that you recognize how important metastasis is in understanding lung cancer fatalities. We’ve highlighted this point throughout the review, for example on lines 57-5858, 61-62, 111, 219, and 465 with relevant references.
For patients diagnosed with late-stage lung cancer, especially NSCLC, precision medicine has already made a strict in extending their lives. The models we've examined in this review enrich our understanding of lung cancer and offer new, valuable insights into the complexities of this disease's metastasis.
Comment 2: - Please consider how precision medicine can overcome the chemotherapy and radiotherapy resistance in lung cancer or how it can help to improve the patient response.
Response: We agree and have followed this theme in our revision adding the clinical implications and examples after reviewing each model. Please see Tables 2 and 3 for the summary.
Comment 3: - The current topic could improve the overall outcome of research approach in lung cancer and also could be first steps to decrease the mortality rate in this specific type of cancer. How the authors see the precision medicine and personalized medicine will benefit the patients for this approach.
Response: We agree and have revised our manuscript significantly. Please see the response to the above in Comment 2.
Comment 4: - How does the application of precision medicine will improve the health policy in future in this field?
Response: We appreciate the notion of this important point. However, the impact of precision medicine on health policy is a complexed topic that can be influenced by many factors such as gender, ethnicity, socioeconomic status, regions and countries. It is beyond the scope of this manuscript as it is currently at 32 pages with 198 references.
Comment 5: - Please provide a table compare advantage and disadvantage of each model they discussed in the paper.
Response: Thank you for your suggestions. We have added Table 2 to compare the advantages and disadvantages of in vitro models and Table 3 to compare in vivo models.
Round 2
Reviewer 2 Report
Comments and Suggestions for Authors
The authors reasonably addressed all the points the reviewer suggested.
Reviewer 4 Report
Comments and Suggestions for Authors
I am very pleased with the high quality revision.